# Genetic Diversity and Spatiotemporal Dynamics of Chikungunya Infections in Mexico during the Outbreak of 2014–2016

**DOI:** 10.3390/v14010070

**Published:** 2021-12-31

**Authors:** Eduardo D. Rodríguez-Aguilar, Jesús Martínez-Barnetche, Cesar R. González-Bonilla, Juan M. Tellez-Sosa, Rocío Argotte-Ramos, Mario H. Rodríguez

**Affiliations:** 1Centro de Investigación Sobre Enfermedades Infecciosas, Instituto Nacional de Salud Pública, Av. Universidad 655, Cuernavaca 62100, Mexico; danielro_17@hotmail.com (E.D.R.-A.); jmbarnet@insp.mx (J.M.-B.); jmtellez@insp.mx (J.M.T.-S.); rsargo@insp.mx (R.A.-R.); 2Instituto Mexicano del Seguro Social, Mexico City 07760, Mexico; cesar.gonzalezb@imss.gob.mx

**Keywords:** chikungunya virus, genetic variability, mutations, combined phylogenetic analysis, phylogeography

## Abstract

Chikungunya virus (CHIKV) is an alphavirus transmitted by *Aedes* mosquitoes, which causes Chikungunya fever. Three CHIKV genotypes have been identified: West African, East-Central-South African and Asian. In 2014, CHIKV was detected for the first time in Mexico, accumulating 13,569 confirmed cases in the following three years. Studies on the molecular diversification of CHIKV in Mexico focused on limited geographic regions or investigated only one structural gene of the virus. To describe the dynamics of this outbreak, we analyzed 309 serum samples from CHIKV acute clinical cases from 15 Mexican states. Partial NSP3, E1, and E2 genes were sequenced, mutations were identified, and their genetic variability was estimated. The evolutionary relationship with CHIKV sequences sampled globally were analyzed. Our sequences grouped with the Asian genotype within the Caribbean lineage, suggesting that the Asian was the only circulating genotype during the outbreak. Three non-synonymous mutations (E2 S248F and NSP3 A437T and L451F) were present in our sequences, which were also identified in sequences of the Caribbean lineage and in one Philippine sequence. Based on the phylogeographic analysis, the viral spread was reconstructed, suggesting that after the introduction through the Mexican southern border (Chiapas), CHIKV dispersed to neighboring states before reaching the center and north of the country through the Pacific Ocean states and Quintana Roo. This is the first viral phylogeographic reconstruction in Mexico characterizing the CHIKV outbreak across the country.

## 1. Introduction

Chikungunya virus (CHIKV) is an alphavirus transmitted by the bite of infected female *Aedes aegypti* and *Aedes albopictus* mosquitoes. It causes chikungunya fever (CHIKF), an acute febrile disease with four clinical forms: acute, atypical acute, severe acute and chronic [1]. The distinctive clinical features are usually fever and polyarthralgia, often accompanied by polyarthritis. However, myalgia, headache, rash, fatigue, diarrhea and oedema can also occur [2]. Atypical acute CHIKF includes neurological, cardiovascular, skin, renal, and respiratory manifestations [3,4,5]. Cardiac or multiple organ failure are prevalent in severe acute cases [6]. CHIKV contains a 12 kb positive-sense, single-stranded RNA genome coding nonstructural proteins (nsP 1 to 4) at the 5′ end, and structural proteins (C, E3, E2, 6K, and E1) at the 3′ end [7]. Three distinct evolutionary genotypes have been identified: the West African, East-Central-South African (ECSA), and Asian CHIKV [8].

The first cases of CHIKF were described in Tanganyika (present-day Tanzania) during the 1950s [9]. These were followed by some sporadic outbreaks in Africa and Asia [10]. Since 2004, CHIKV exploded onto the global scene as a major emerging pathogen in several outbreaks that infected up to 10 million people [2]. Phylogenetic studies revealed that these outbreaks were associated with three independent CHIKV lineages [11,12,13,14]. The major CHIKF outbreaks were caused by virus strains of the Indian Ocean lineage (IOL), which evolved from the ECSA genotype [15]. Adaptive mutations E1:A226V [11] and E2:L210Q [16] which facilitate their transmission by *Ae. albopictus,* were related to this lineage. The IOL emerged in Kenya in 2004 and spread to islands in the Indian Ocean and Southeast Asia [17]. A CHIKV strain of the IOL, presumably transported from India by an infected traveler, was responsible for a CHIKV outbreak in Italy in 2007 [18]. A second series of outbreaks, caused by phylogenetically distinct CHIKV strains, also belonging to the ECSA genotype, began in 2006 in Cameroon and spread to Gabon in 2007 [14]. A third CHIKV lineage was responsible for a 2006 outbreak in Malaysia and belonged to the original endemic Asian genotype [19].

The first autochthonous cases of CHIKF in the Western Hemisphere were reported in 2013 in the Caribbean region at St. Martin [20]. These were caused by an Asian lineage strain, apparently imported from Southeast Asia or Oceania [21]. After its dissemination through the Caribbean islands, this strain spread through all the Central American countries, most of South America, and northwards into Mexico [22]. The first autochthonous case of CHIKF in Mexico was reported in mid-October 2014 in the city of Arriaga, in the southern state of Chiapas [23]. CHIKF cases were also detected during the first two weeks of October 2014 in Ciudad Hidalgo, in southern Chiapas [24]. By early 2015, CHIKF cases began to appear in the neighboring states of Oaxaca and Guerrero [25]. In 2015, 11,577 cases were reported in the country. In the following years reported cases decreased considerably, 757 cases were reported in 23 states during 2016 and 61 cases in 14 states in 2017.

A study, at the beginning of the outbreak in Mexico, classified the circulating strain into the Asian genotype which was phylogenetically grouped with lineages of the Caribbean strains [26]. Most of the studies on the molecular diversification of CHIKV in Mexico focused on limited geographic regions of the country, generally at the borders of Chiapas [25] and Tamaulipas [27] or investigated only the virus structural E1 gene [28,29].

Although the transmission and spread of CHIKV are now considered a moderate risk, the threat of new outbreaks is not ruled out due to the high infestation of *Aedes* mosquitoes in the country [30]. This study advances current knowledge of the spatiotemporal dispersion of CHIKV outbreak in Mexico, using the phylogenetic information, obtained from structural and non-structural genes.

## 2. Materials and Methods

### 2.1. Virus Isolation, Reverse Transcription-Polymerase Chain Reaction Amplification, and Nucleotide Sequencing

A total of 309 CHIKV serum samples from 15 Mexican states (Table 1), obtained between 2015 and 2016, were provided by the central laboratory of epidemiology of the Mexican Social Security Institute (IMSS).

The use of hypervariable regions to elucidate phylogenetic relationships at low taxonomic levels has proven successful [31,32]. For the present phylogenetic analysis, we chose to sequence three hypervariable regions of the CHIKV genome. These regions were selected through a genetic variability analysis of 368 whole genome aligned sequences from the Virus Pathogen Database and Analysis Resource (ViPR) of the National Institute of Allergy and Infectious Diseases (NIH/DHHS). To identify regions with the greatest variability, the number of nucleotide changes and the ratio of synonymous and non-synonymous mutations were determined for each codon in forty-five non-overlapping nucleotide-sliding windows. The regions with the highest number of mutations and highest ratios of synonymous and non-synonymous mutations (the amino terminal region from the NSP3 gene, a central region comprising part of the domains I and II of the E1 gene, and the B and C domains of the E2 gene) were chosen for subsequent analysis (Appendix A).

The preparation of CHIKV sequence libraries from serum samples followed the instructions of Preparing 16S Ribosomal RNA Gene Amplicons for the Illumina MiSeq System workflow [33]. Viral RNA was extracted from 140 µL of serum using QIAamp Viral RNA Mini Kits (Qiagen, Germany) and the genome regions were amplified using primers containing overhang adapters in a Triplex one-step PCR reaction (Appendix A). Reverse transcription was performed using Superscript III Reverse Transcriptase (Invitrogen) and amplified by polymerase chain reaction (PCR) using Platinum Taq DNA polymerase (Invitrogen) (For primers and PCR conditions see Appendix A). The three amplicons corresponding to NSP3 (514 nt), E2 (547 nt) and E1 (479 nt) amplified regions were visualized in 2% agarose gels (Appendix A) and subsequently purified using the Agencourt AMPure Beads^®^ system (Beckman Coulter, (United States, Indiana, Indianapolis)). Biotinylated magnetic AMPure beads allow for selection of specified cDNA products bound to streptavidin. 50 μL of amplified cDNA from samples were mixed and purified two times with AMPure XP beads at a 1.8:1 ratio (beads:sample), this ratio allows for optimal selection of all products higher than 100 nt. The three amplicons of each sample were pooled according to the location and date of sampling in 24 pools of a maximum of 20 samples for each library (Appendix A) The Illumina system adapters containing indexes (known short sequences used to identify the sequences belonging to each library) were added using PCR reactions (Appendix A) (For index and adapters see Appendix A). The libraries were sequenced in the MiSeq platform (Illumina, (United States, California, San Diego)), at the Mexican National Institute of Genomic Medicine (INMEGEN, Mexico, CDMX, Mexico City), with a 600 cycles V3 kit with a paired-end sequencing configuration, to obtain 300 bp paired end overlapping reads, following the manufacturer’s instructions (Appendix A).

### 2.2. Data Sets

Adapters and poor quality reads (minimum Phred quality of 30) were removed using Trimmomatic software (Version 0.32, USADELLAB, Aachen, Alemania) [34]. The remaining sequences were aligned to the reference genome NC_004162.2 using Bowtie2 software (version 2.3.4.1, SOURCEFORGE, San Diego, California) [35]. Each data set was trimmed to a common length and the sequences of the regions of NSP3, E1, and E2 genes of each sample were concatenated in a single sequence of 1341 nucleotides, and the amplicon sequence variants (ASVs) within each pool were determined by clustering centroids with 100% identity using the VSEARCH software (Version 2.13.6, UNINETT, Oslo, Norway) [36] (Appendix A). The variants with the highest number of readings in each pool were accumulated until reaching an effective sample depth of 95% [37]. To prevent overrepresentation of some locations and dates, we constructed a preliminary MCC tree (Appendix A), sequences with the same location and date that were grouped in the same clade were collapsed into a single sequence.

### 2.3. Phylogenetic Analysis

As a pre-processing step, sequence recombination was screened using GARD (Genetic Algorithm for Recombination Detection) [38] available in the Datamonkey web server. Clustal Omega was used for multiple sequence alignment based on the Percent nucleotide identities (PNI) calculated using p-distances [39]. The best-fit model of nucleotide substitution was selected based on the Bayesian Information Criterion (BIC) available in ModelTest 3.5 [40]. The GTR + G + I model (general time-reversible model with gamma-distributed rates of variation among sites and a proportion of invariable sites) was found to be the best-fit model.

The temporal information of the sequence data was used to estimate the evolutionary rate and the time to the most recent common ancestor (TMRCA), by generating a MCC tree, using the Bayesian Markov Chain Monte Carlo approach, as implemented in BEAST2 2.6.4 [41]. For this, we employed both strict and relaxed (uncorrelated exponential and uncorrelated lognormal) clock [42] models with the Bayesian Skyline tree prior. Three independent runs of the Bayesian Markov Chain Monte Carlo were carried out, each with at least 100 million generations and a sampling frequency of 10,000. The posterior probability and marginal likelihood of the models were used to choose the most suitable model for the data [43]. Tracer 1.5 assessed the convergence of the chain and the MCC tree was visualized in FigTree (Version 1.2.3, University of Edinburgh, Edinburgh, United Kingdom). The Bayes Factor analysis indicated that the uncorrelated exponential clock model fitted better than the strict clock or uncorrelated lognormal clock model. The corresponding output files generated by BEAST were utilized for further analysis.

### 2.4. Phylogeographic Inference

Phylogeny-trait association tests (AI, Association Index, and PS, Parsimony Score) available in BaTS [44] were performed to evaluate the association between phylogeny and geographical locations of the sequence data and hence the suitability for phylogeographic analysis. The rate of nucleotide substitution per site, per year (subs/site/year), the time to the most recent common ancestor (TMRCA), and the spatial diffusion rates (i.e., the rate at which viral lineages move among sampled locations) were jointly estimated from the date and location of each CHIKV sequence. For this, we used a Bayesian approach implemented in the Markov chain Monte Carlo (MCMC) inference framework of the BEAST v2.6.5 software package [45]. Analyses were carried out using a general time reversible (GTR) model with a discretized gamma-distributed across-site rate variation (GTR + I + Γ4) substitution model and a relaxed uncorrelated lognormal molecular clock model. MCMC was run sufficiently enough to ensure stationarity. The convergence of parameters was assessed by calculating the Effective Sample Size (ESS) using TRACER [46]. Maximum clade credibility (MCC) trees were summarized using TreeAnnotator v1.8 and visualized with FigTree v1.4.2. An MCC tree is a point-estimate characterizing the posterior distribution of trees and represents the tree topology yielding the highest posterior probabilities of individual clades. The branch lengths in these MCC trees are posterior median estimates [47]. Further, the tree nodes were annotated with their most probable (modal) location via color labeling. The MCC tree obtained was the input to the program SPREAD 1.0.3 [48] to visualize and analyze the dispersion pathways.

## 3. Results

### 3.1. Nonsynonymous Mutations in Mexican CHIKV Sequences

We obtained between 711 and 5228 different concatenated sequences per pool, corresponding to the NSP3, E2, and E1 partial genes. After choosing the representative sequences from each pool (the sequences with the highest number of reads), we obtained 238 new sequence variants (Between 5 and 19 new sequence variants for each pool). After collapsing the sequences with the same location and date that were grouped within the same node according to the preliminary MCC tree (Appendix A), 59 variant sequences remained. Eleven were isolated from patients living in the states of Baja California Sur, 2 from Baja California, 5 from Chiapas, 9 from Colima, 3 from Mexico City, 1 from Guerrero, 1 from Mexico State, 2 from Michoacan, 2 from Nuevo Leon, 6 from Oaxaca, 3 from Quintana Roo, 3 from Sinaloa, 3 from Tabasco, 6 from Veracruz, and two from Yucatan (Appendix A).

The nucleotide and amino acid identity of our sequences were 99.50–100% and 99.35–100%, respectively. Compared to the first Mexican sequences reported in October 2014 [23], our sequences depicted 45 mutations in the NSP3 gene region (22 synonymous and 23 non-synonymous), 44 in E1 (20 synonymous and 24 non-synonymous), and 42 in E2 (19 synonymous and 23 non-synonymous). All non-synonymous mutations were unique to each isolate (Appendix A and Appendix A). Compared to the ECSA genotype, our sequences depicted 14, 7, and 4 non-synonymous mutations in the proteins NSP3, E2, and E1, respectively. Compared to the Asian genotype, four non-synonymous mutations in protein NSP3 and one non-synonymous mutation in protein E2 were identified. Three non-synonymous mutations (E2 S248F and NSP3 A437T and L451F) were present in our sequences, the Caribbean lineage and one Philippine sequence (Table 2).

### 3.2. Phylogenetic Analysis

We did not find recombinant sequences in the pre-processing phylogenetic analysis step. To assess the evolutionary relationship between our 59 new sequences and CHIKV sequences sampled globally, we aligned our sequences with those from four of the first cases detected in Mexico [23] and 48 CHIKV ECSA and Asian genotypes sequences obtained from GenBank. The relationship of the CHIKV sequences was examined using a MCC tree reconstructed with concatenated NSP3, E2 and E1 partial genes (1341 nucleotides). Our phylogenetic analysis resulted in a MCC tree that resolved the ECSA and Asian genotypes with a posterior probability of 1, the IOL clade with posterior probability of 1 and the Caribbean epidemic lineage with posterior probability of 1 (Figure 1). Isolates from the Indian Ocean and Caribbean outbreaks conformed monophyletic clades in the ECSA and Asian genotypes, respectively. Our 59 new sequences clustered in three different clades within the Caribbean lineage of the Asian genotype. Other sequences reported from other countries in the Caribbean, Central and South America, from 2013 to 2015, were in the same cluster. These were closely grouped with one sequence of Saint Martin sampled at the beginning of the outbreak in the Americas in 2013 (posterior probability of 1) and one Philippine sequence from 2013 (posterior probability of 1) (Figure 1).

### 3.3. Phylogeographic Inference

The phylogeny-trait association tests indicated a strong association between sampling location and phylogeny, supporting the suitability of phylogeographic analysis. Among the locations analyzed, coast locations (B.C.S., Chiapas, Colima, Guerrero, Oaxaca, Quintana Roo and Veracruz) showed stronger phylogenetic clustering (*p*-value = 0.01) (Appendix A).

To investigate the relationship among our sequences, we used a maximum clade credibility approach. The estimated substitution rate was 2.3 × 10^−3^ substitutions per site per year (95% higher posterior density (HPD): 1.44–3.1 × 10^−3^ subs/site/yr.). The estimated date of the most recent common ancestor (tMRCA) was January 2014, with a 95% HPD between April 2012 and August 2014.

The maximum clade credibility phylogeographic tree divided the Mexican sequences into three main clades. In the first (Clade A), sequences from the south of the country (Tabasco, Quintana Roo, Chiapas, Oaxaca, and Veracruz) and the North Pacific (Sinaloa and Baja California Sur) were grouped together. The most probable root state location for this node was one strain obtained in Quintana Roo (location probability of 0.69). The Baja California Sur sequences were grouped into a single clade, from which a Nuevo León sequence emerged (location probability 0.76) (Figure 2).

The second clade (Clade B) grouped sequences from the Pacific coast (Guerrero, Colima, Michoacan, Sinaloa, Baja California, and Baja California Sur), center (Mexico City and Mexico State) and north (Nuevo Leon and Tamaulipas) of the country. The most probable root state location for this node was one strain obtained in southeast Chiapas in October 2014 at the beginning of the outbreak (location probability of 1). The next internal nodes indicated Guerrero and Colima as the most probable locations (location probability of 0.51). After this, the descending sequences were grouped into two clades: one grouping sequences from the north Pacific coast (Michoacan, Sinaloa, and BCS) and the other grouping sequences from the center of the country (Mexico State and Mexico City) along with one sequence from Tamaulipas (Figure 2).

The third clade (Clade C) grouped sequences from the south of the country (Yucatan, Tabasco, Quintana Roo, Chiapas, Oaxaca, and Veracruz) and the North Pacific (Colima, Sinaloa and Baja California Sur). One sequence which was the most probable root state location for this clade (location probability of 0.67) was obtained in southern Chiapas, in November 2014 at the beginning of the outbreak [24]. The clade was divided into two groups: the first grouping sequences from Chiapas, Tabasco, and Yucatán with the most probable root location in Quintana Roo (location probability 0.67), and the second with the most probable root location in Oaxaca (location probability 1.0) grouping sequences from Chiapas, Veracruz, Colima, Sinaloa, and Baja California Sur. The Jalisco sequence belonging to an imported case occurred in May 2014 [23] grouped in Clade C. However, it did not spread to other locations (location probability of 1.0) (Figure 2).

Transformed data from the maximum clade credibility phylogeographic into a phylogeographic model, suggested that CHIKV spread from Chiapas to Oaxaca, Guerrero, and Quintana Roo during the first three months of 2015 (Figure 3a). By June 2015, the virus spread through the south and center of the country, reaching Veracruz, Mexico State, and Mexico City from Quintana Roo and went along the Pacific coast to Colima (Figure 3b). At the end of 2015, the virus reached the north of the country in Sinaloa, Baja California, Baja California Sur, and Tamaulipas (Figure 3c). Finally, by September 2016 (most recent samples) the virus circulated from north to south in the country (Figure 3d).

## 4. Discussion

Previous studies on the molecular diversification of CHIKV in Mexico were restricted to limited geographic regions of the country, such as Chiapas [25] and Tamaulipas [27]. Herein we extended these studies with samples collected across Mexico.

The standard approach, using only one structural gene of the virus [26,29], to study molecular diversification has been shaped by the relative technical easiness for increasing the number of taxa and the desire to study as many taxa as possible. However, the use of small amounts of phylogenetic signal generates phylogenetic hypotheses that are incongruent or lack support [56]. For instance, a maximum likelihood analysis that used a 1044 bp region from the E1 gene did not resolve the ECSA lineage as monophyletic [29], as has been shown in whole genome analyses [57]. As previously observed using other approaches, combined gene analyses can be superior to single-gene analyses for the resolution of internal branches and the position of taxa forming long branches [58]. Here, the sequence constructed with concatenated highly variable regions the NSP3, E1 and E2 proved to resolve the clades of the ECSA and Asiatic genotypes as monophyletic, including the IOL and Caribbean lineages.

In this study, we used 59 sequences, along with published sequences, from CHIK cases obtained during the outbreak that occurred in Mexico between 2014 and 2016. We found 70 non-synonymous mutations in our new sequences compared to the sequences sampled at the beginning of the outbreak, 23 in NSP3, 24 in E1, and 23 in E2. However, we did not observe the same mutations in isolates from different geographical regions or times, suggesting that diversifying selection does not occur. The mutation E1 A226V found in IOL-ECSA sequences that confers increased infectivity to *Ae. albopictus* and favors the dissemination of CHIKV but has inconsistent effects in the infectivity to *Ae. Aegypti* [59,60], was not present in our samples. However, the E1 K211E mutation, occurring in all our samples, was reported to increase virus fitness in *Ae. aegypti*, increasing the virus infectivity, dissemination and transmission, compared to the E1 A226V virus [61]. In the same way, three non-synonymous mutations (E2 S248F and NSP3 A437T and L451F) were present in all of our sequences, as well as in the sequences from the Caribbean outbreak [62], pointing at this geographic region as the origin of the virus that caused the Mexican outbreak, as previously inferred [26].

A recent study demonstrated that another mutation in this site (E2 S248L) was beneficial for the dissemination of the virus by *Ae. albopictus* [55]. Whether the amino acid substitution of E2 S248F would have a beneficial effect on the viral fitness in *Ae. aegypti*, the main mosquito vector in Mexico and the Caribbean region needs further investigation. There is not much information about the A437T and L451F NSP3 mutations. The A437V mutation (same site as the A437T mutation) has been reported in samples of the ECSA genotype [51], suggesting that amino acid changes at this site may be related to adaptations to *Ae aegypti* and *Ae albopictus*, as could be the case of the E2 S248F mutation.

Another study found these three mutations (E2 S248F and NSP3 A437T and L451F) in samples from the CHIKV outbreak in the Philippines in 2012 [54]. This study suggests the possibility that the strain called Cosmopolitan Asian CHIKV (CACV) responsible for outbreaks in the Caribbean had its origin in the Philippines. Our results support this hypothesis, as these mutations were present only in the Caribbean lineage (including our sequences) and one isolate from the Philippines. In addition, in our phylogenetic analysis, the sequence from the Philippines clustered closely to the clade of the Caribbean lineage.

Before assessing the spread dynamics of the sequences included in our study, we performed a phylogeny-trait association test to estimate the association between sampling location and phylogeny. Although the values for PS and AI were significant in general (*p*-value < 0.001), locations with the larger number of samples showed greater association (Appendix A). A possible explanation for these results is that discrete trait analysis is sensitive to the relative sampling intensity of subpopulations, such that the sampling strategy can influence the results, particularly when migration rates are high [63]. Unbiased sampling can be very hard to achieve, as it requires knowledge of the full geographic range of an outbreak, access to the entire range, and extensive sampling and sequencing efforts. As our samples were obtained from clinical cases demanding attention, sampling size at different locations was not controlled and the results should be assessed with this limitation in mind.

To assess the spread dynamics of CHIKV lineages in Mexico, we analyzed the NSP3, E2, and E1 partial genes of our 59 sequences along with other sequences sampled globally. Our phylogenetic analysis resolved the genotypes ECSA and Asian with good branch support for all lineages including the IOL and Caribbean lineage. Our isolates were closely grouped with Central American sequences within the Asian genotype, more precisely in the Caribbean lineage, as reported in previous studies [23,25,26,29]. This supports the hypothesis that CHIKV reached Mexico through the Caribbean and Central America [26], where outbreaks were reported before the outbreak began in Mexico [64]. The CHIKV epidemic in the Americas derived from the re-emergent Asian lineage [21]. According to our results, the ECSA genotype previously reported in Brazil [65] was not found and the only circulating genotype during the outbreak in Mexico was of the Asian lineage. The closeness of clustering among the sequences from Central America and Mexico suggests the possibility of epidemiologically related transmission between Mexico and Central America.

A previous study documented that samples obtained in 2014 in the southeast and southern Chiapas grouped with other isolates from Nicaragua and the Caribbean [66]. According to our phylogenetic and phylogeographic analysis, the sequence isolated in southeast Chiapas (15 October 2014) could have moved through Mexico until reaching the northern border in November 2015 (Clade B), while the sequence isolated in southern Chiapas a month later (15 November 2014) spread throughout the south of the country and the Pacific coast (Clade C).

A previous study hypothesized that there was at least another introduction of CHIKV to the country, since five of their sequences from southern Chiapas grouped with sequences from Nicaragua that were obtained during 2014 and 2015 [25]. Our results also suggest a third introduction, since our sequences clustered closely with sequences from Central and South America in three different clades. In addition, sequences collected in Chiapas did not cluster with the sequences obtained early in the outbreak (two from Chiapas and one from Jalisco) and formed an independent clade with sequences from other parts of the country (Clade A).

The one sample from Jalisco isolated from an imported case at the beginning of the outbreak [23] grouped in Clade C. However, the most probable root location for his clade (Location probability of 0.67) was a sequence obtained in southern Chiapas at the beginning of the outbreak, suggesting that this introduction from an imported case did not spread throughout the country or did so in a limited way.

This is the first spatiotemporal reconstruction of the evolutionary history of CHIKV across Mexico during the outbreak occurring between 2014 and 2016. CHIKV spread from the southern border in Chiapas through the Mexican Caribbean and the Pacific states, before reaching the center and north of the country. The main dissemination of the virus occurred from the south to the rest of the country. Human movements occur mainly between neighboring states, but Quintana Roo, with high local and international tourist and job attractions, possibly played a major role in the dissemination of this virus on a seasonal basis.

## 5. Conclusions

Phylogenetic information from a combined analysis of structural and non-structural genes can offer higher resolution than a region of the same size from a single gene such as E1.

The unique amino acid substitutions common among our samples and samples from other parts of the world suggest that mutations that increase the ability of the virus to replicate and spread in areas where the main vector is *Ae. aegypti* occurred before its introduction into Mexico. While the possible biological importance of these mutations is still unknown, the genetic signatures identified in the study represent interesting candidates for future in-depth study and epidemiological follow-up.

The phylogenetic and phylogeographic analyzes indicated that the only circulating genotype was of the Asian lineage. Multiple introductions of the virus to southern Mexico from Central America and the Caribbean spread in a short and localized way, but the state of Quintana Roo may have played a major role in the spread of the virus to other regions of the country.

## Figures and Tables

**Figure 1 viruses-14-00070-f001:**
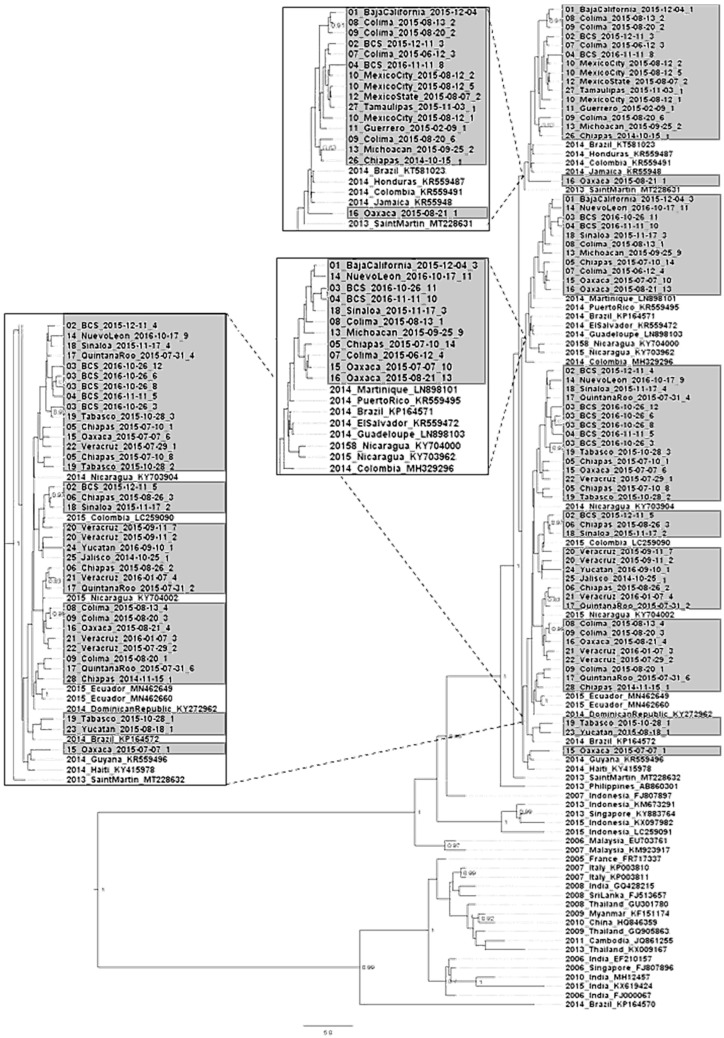
Maximum clade credibility phylogeny constructed with the 59 CHIKV sequences collected in this study along with the four sequences obtained at the beginning of the outbreak and 48 sequences from other parts of the world. Taxon labels include accession number, isolation place, and year. The sequences collected in this study are shaded in grey. BPP values are shown for relevant nodes. The three clades identified in this study are magnified within bounded rectangles.

**Figure 2 viruses-14-00070-f002:**
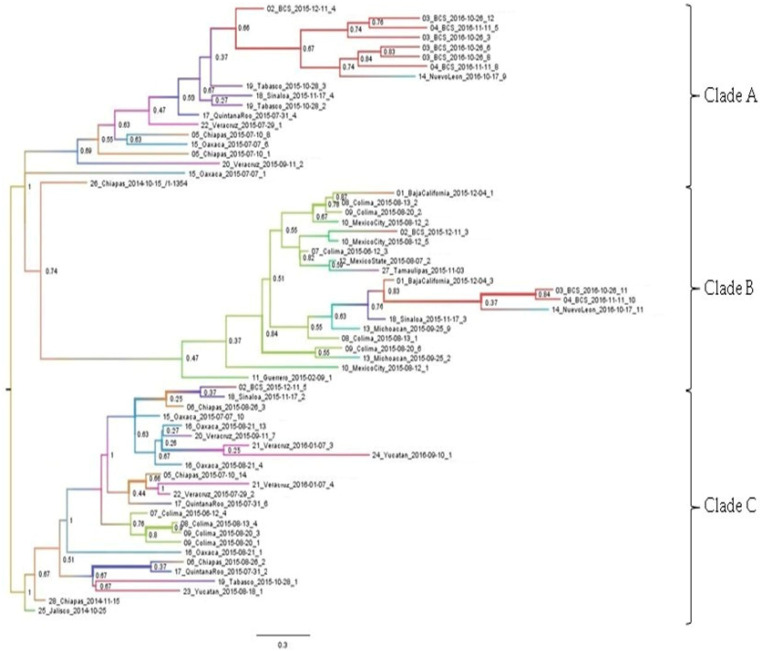
Phylogeography of CHIKV during the outbreak 2014–2016 in Mexico. Maximum clade credibility phylogeographic tree of the Mexican CHIKV outbreak clade. Branch colors represent most probable inferred locations. The branch thicknesses are sized in proportion to root location probability.

**Figure 3 viruses-14-00070-f003:**
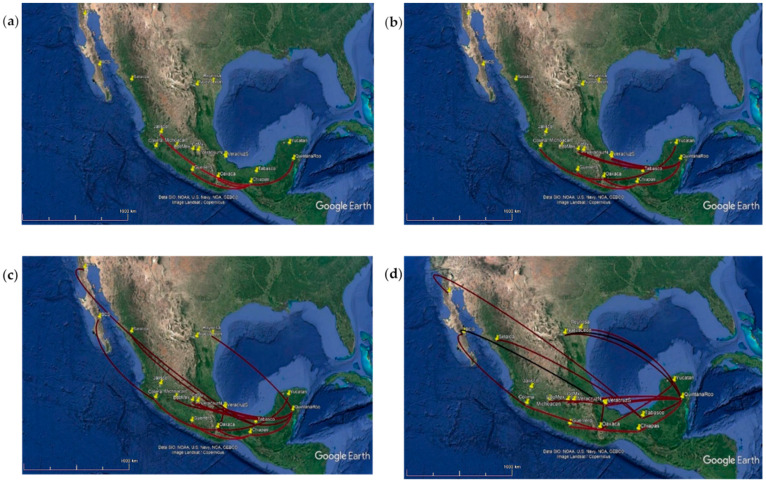
Spatiotemporal circulation of CHIKV during the outbreak 2014–2016 in Mexico represented with snapshots of the dispersal pattern for (**a**) December 2014, (**b**) June 2015, (**c**) December 2015 and (**d**) September 2016. Lines represent MCC phylogeny branches projected on the surface.

**Table 1 viruses-14-00070-t001:** Numbers and location origins of serum samples used in this study.

Location	2015	2016	Total
Baja California (B.C.)	5		5
Baja California Sur (B.C.S.)	14	24	38
Chiapas	23		23
Colima	39		39
Ciudad de Mexico (CdMx)	3		3
Guerrero	16		16
Estado de Mexico (EdoMex)	9	1	10
Michoacan	16	1	17
Nuevo Leon		4	4
Oaxaca	41		41
Quintana Roo	15	2	17
Sinaloa	9	2	11
Tabasco	5		5
Veracruz	28	29	57
Yucatan	21	2	23
Total	244	65	309

**Table 2 viruses-14-00070-t002:** Amino Acid Changes Identified among ECSA, IOL, Asian, Caribbean and Mexican CHIKV sequences.

Gene	Mutation	Genotype/Lineage					Effect	References
		ECSA	IOL	Asian	Caribbean	Mexico		
E1	A226V	A	V	A	A	A	Leads to increased fitness, dissemination to the salivary glands and transmissibility of the virus by *Aedes albopictus*	Schuffenecker et al., 2006 [11]
	K211E	E	K	E	E	E	Increases fitness for *Ae. aegypti*, increase in virus infectivity (13 fold), dissemination (15 fold) and transmission (62 fold) compared to E1:226A virus.	Shrinet et al., 2012 [49]
	T145A	T	T	A	A	A	Unknown	Tandel et al., 2019 [50]
	S225A	A	A	S	S	S		I-Ching Sam et al., 2012 [51]
E2	G118S	G	G	S	S	S	Unknown	Chong-Long Chua et al., 2016 [52]
	S194G	G	G	S	S	S		
	V255I	I	I	V	V	V		
	D205G	G	G	D	D	D	Unknown	O. Suhana et al., 2019 [53]
	S207N	N	N	S	S	S		
	S248L	L	L	S	F *	F *		I-Ching Sam et al., 2012 [51]/*Kim-Kee Tan et al., 2015 [54]
	K252Q	Q	K	K	K	K	Increased adaptation to *A. albopictus*.	Konstantin A. Tsetsarkin et al., 2014 [55]
NSP3	I383T	T	T	I	T	T	Unknown	O. Suhana et al., 2019 [53]
	I413T	T	T	I	I	I		
	Q434L	L	L	Q	Q	Q		
	A437V	V	V	A	T	T		Kim-Kee Tan et al., 2015 [54]
	I449M	M	M	I	I	I		I-Ching Sam et al., 2012 [51]
	L451F	L	L	L	F	F		Kim-Kee Tan et al., 2015 [54]
	R452Q	Q	Q	R	R	R		I-Ching Sam et al., 2012 [51]
	I457T	T	T	I	I	I		
	T458A	A	A	T	T	T		
	V459T	T	T	V	V	V		
	L461P	P	P	L	L	L		
	S462N	N	N	S	S	S		
	P471S	S	S	P	P	P		
	D483N	N	N	D	N	N		Kim-Kee Tan et al., 2015 [54]
	D484E	E	E	D	D	D		I-Ching Sam et al., 2012 [51]

## Data Availability

The data presented in this study are openly available in GenBank (SRA) with accession number PRJNA772723.

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
