# Peer review of "Genetic Diversity and Spatiotemporal Dynamics of Chikungunya Infections in Mexico during the Outbreak of 2014–2016"

_viruses, 2021, doi:10.3390/v14010070_

Round 1
Reviewer 1 Report
Although this paper includes a lot of new sequencing data there are no whole genome sequences. In addition, the data is not entirely clear and is quite confusingly presented. The alleged novelty (3.1, Table 2 is hard to find). The 70 aa changes are not well described and its difficult to follow whether they are actually all unique to single isolates – and thus may represent sequencing errors or random isolations of quasispecies variants. New insights into CHIKV evolution are hard to extract from this paper.
Major issues
Introduction
L36 The key disease manifestations are polyarthralgia and polyarthritis, fever and rash (headache is a minor manifestation) – please describe better (Nat Rev Rheumatol. 2019 Oct;15(10):597-611).
Results
L182 3.1 says Novel but quite unclear in Table 2 what is deemed novel – in fact I cannot find any novel aa changes at all? The Mexican isolates are ALL the same as the Caribbean, from whence they are likely derived.
Discussion
“molecular diversification of CHIKV in Mexico were restricted to specific geographic regions of the country, such as Chiapas [21] and Tamaulipas [23].” This seems a little odd unless followed by “whereas herein we studied sequences from across Mexico” or similar.
Seems odd to criticise others for using partial sequences when the paper itself also uses partial sequences. Whole genomes would clearly be ideal.
“We found 70 non-synonymous mutations in our new sequences” unclear mutations away from what baseline sequence – Caribbean? Or are these differences between all the viruses sequenced in the paper. Are these described in Table 2?
“ we did not find the same mutation in isolates from different geographical regions or times, suggesting that that diversifying selection does not occur” very unclear – which mutation is this is referring too. Our does this mean of the 70 mutations each mutation was only found once and then only in Mexican viruses sequnced in this study? If so does this not suggest sequencing errors or randome quasispecies sampling? Also unclear what is meant here by diversifying selection. Random mutations that have had no chance to be selected? Not really selecting for diversity, instead there is no fixation of the mutation?
Fig. 1 is difficult to read or understand and appears truncated in the middle somehow.
Fig. 3 MCC phylogeny branches projected onto surface - very unclear. Do any of the red lines somehow relate to the clades in Fig. 2?
Minor issues
L14 next – following
L 16 not sure that the sequencing provided in this paper provides information on dynamics.
L22 only is unclear here – why only ? Perhaps remove the word only?
L44 This references is quite dated and the number of cases is >10 million (ibid)
L201 “None of the non-synonymous mutations was found in more than one sequence “ this needs rephrasing “all non-synonymous mutations were unique to each isolate”?
L207 Again “only” is used – quite unclear.
L217 Not clear what a pre-analytical phylogenetic analysis might be.
Author Response
Reviewer 1
Reviewer: Although this paper includes a lot of new sequencing data there are no whole genome sequences.
Answer: Thank you for this reviewer’s observation, although the use of whole genomes is ideal, the construction of whole genomes would have severely restricted the number of samples sequenced. The use of a combined gene analyzes has been shown to be superior to single-gene analyzes. It allowed us to include a large number of samples increasing the probability to find new variants. We supported the use of hypervariable regions: Line96-97 The use of hypervariable regions to elucidate phylogenetic relationships at low taxonomic levels has proven successful [31,32].
- Yang, B.; Wang, Y.; Qian, P.-Y. Sensitivity and Correlation of Hypervariable Regions in 16S RRNA Genes in Phylogenetic Analysis. BMC Bioinformatics 2016, 17, 135, doi:10.1186/s12859-016-0992-y.
- Dong, W.; Liu, J.; Yu, J.; Wang, L.; Zhou, S. Highly Variable Chloroplast Markers for Evaluating Plant Phylogeny at Low Taxonomic Levels and for DNA Barcoding. PLOS ONE 2012, 7, e35071, doi: 10.1371/journal.pone.0035071.
Reviewer: In addition, the data is not entirely clear and is quite confusingly presented. The alleged novelty (3.1, Table 2 is hard to find).
Answer: Thank you for this reviewer´s observation, we have included more data in the manuscript and supplements. Novelty is no longer alleged as to the evolutionary history of CHIKV and table 2 was restructured
Reviewer: The 70 aa changes are not well described and it’s difficult to follow whether they are all unique to single isolates – and thus may represent sequencing errors or random isolations of quasispecies variants.
Answer: Thank you for this reviewer´s observation, we add a new figure (Figure S5) to describe these mutations. This is unlikely to be sequencing errors as we were very strict on quality filtering (minimum Phred quality of 30), clustering (clustering centroids with 100% identity) and selecting variants with high coverage (effective sample depth of 95%).
Reviewer: New insights into CHIKV evolution are hard to extract from this paper.
Answer: The aim of this paper was to describe the possible dissemination of CHIKV during the outbreak 2014-2016 in Mexico. Using a phylogenetic analysis, we propose points of entry and the main sites for dissemination. We could document the Caribbean origin of the virus and documented a single Asian ancestry. While no divergent mutations were detected.
Major issues
Introduction
Reviewer: L36 The key disease manifestations are polyarthralgia and polyarthritis, fever and rash (headache is a minor manifestation) – please describe better (Nat Rev Rheumatol. 2019 Oct;15(10):597-611).
Answer: Thank you for this reviewer´s suggestion, the clinical manifestations of the disease are now described in detail (L 39-45). Now it's written Lines 40-44: “chikungunya fever (CHIKF), an acute febrile disease with four clinical forms: acute, atypical acute, severe acute and chronic. The distinctive clinical features are usually fever and polyarthralgia, often accompanied by polyarthritis. However, myalgia, headache, rash, fatigue, diarrhea and oedema can also occur. Atypical acute CHIKF includes neurological, cardiovascular, skin, renal, and respiratory manifestations. While cardiac or multiple organ failure are prevalent in severe acute cases".
Results
Reviewer: L182 3.1 says Novel but quite unclear in Table 2 what is deemed novel – in fact I cannot find any novel aa changes at all? The Mexican isolates are ALL the same as the Caribbean, from whence they are likely derived.
Answer: Thank you for this reviewer´s observation, certainly there is no novelty in the Mexican sequences compared to the Caribbean sequences (Novelty is no longer alleged as to the evolutionary history of CHIKV). Now it reads: Line195: “Nonsynonymous mutations in CHIKV consensus sequence”. The mutations we found have not been thoroughly investigated. We consider them to be of importance, since the unique amino acid substitutions, common among our samples and samples from the Caribbean suggests they may be related to an increased tability of the virus to replicate and spread in areas where the main vector is Ae. Aegypti.
Discussion
Reviewer: Molecular diversification of CHIKV in Mexico were restricted to specific geographic regions of the country, such as Chiapas [21] and Tamaulipas [23].” This seems a little odd unless followed by “whereas herein we studied sequences from across Mexico” or similar.
Answer: Thank you for the suggestion, the main idea of the paragraph was completed. Now it's written: “Previous studies on the molecular diversification of CHIKV in Mexico were restricted to limited geographic regions of the country, such as Chiapas [25] and Tamaulipas [27] herein we extended these studies with samples collected across Mexico” (L384-386).
Reviewer: Seems odd to criticise others for using partial sequences when the paper itself also uses partial sequences. Whole genomes would clearly be ideal.
Answer: Thank you for this reviewer´s observation, although full genomes would be ideal, the construction of whole genomes would have severely restricted the number of samples sequenced. We propose the use of partial sequences in a combined gene analysis. As previously observed using other approaches, combined gene analyzes can be superior to single-gene analyzes for the resolution of internal branches and the position of taxa forming long branches.
Reviewer: “We found 70 non-synonymous mutations in our new sequences” unclear mutations away from what baseline sequence – Caribbean? Or are these differences between all the viruses sequenced in the paper. Are these described in Table 2?
Answer: Thank you this reviewer´s observation, the baseline sequence was a Mexican sequence sampled at the beginning of the outbreak. Therefore, they are differences between all the viruses sequenced in the paper. Now it's written: (L405-406). “We found 70 non-synonymous mutations in our new sequences compared to the sequences sampled at the beginning of the outbreak” (L405-406). These differences are not presented in Table 2. Figure S5 was added to describe these mutations.
Reviewer: “We did not find the same mutation in isolates from different geographical regions or times, suggesting that that diversifying selection does not occur” very unclear – which mutation is this is referring too. Our does this mean of the 70 mutations each mutation was only found once and then only in Mexican viruses sequenced in this study? If so, does this not suggest sequencing errors or randome quasispecies sampling? Also unclear what is meant here by diversifying selection. Random mutations that have had no chance to be selected? Not really selecting for diversity, instead there is no fixation of the mutation?
Answer: Thank you for this reviewer´s observations, each of the 70 mutations was only found once and only in Mexican viruses sequenced (New figure S5 supplement). This is unlikely to be sequencing errors, as we were strict on quality filtering (minimum Phred quality of 30), clustering (clustering centroids with 100% identity) and selecting variants with high coverage (effective sample depth of 95 %).
Reviewer: Fig. 1 is difficult to read or understand and appears truncated in the middle somehow.
Answer: Thank you for this observation, Description of the figure 1 was improved and data redistributed to increase the font size.
Reviewer: Fig. 3 MCC phylogeny branches projected onto surface - very unclear. Do any of the red lines somehow relate to the clades in Fig. 2?
Answer: Thank you for this reviewer´s observation, Fig. 3 is a representation of the geographic location’s transitions reconstructed in the previous step on a map using geographic coordinates for each location. The red lines represent the 3 clades in Fig. 2. As the three clades share some geographic locations, it is not possible to distinguish among them.
Minor issues
Reviewer: L14 next – following
Answer: Thank you for the suggestion, it is now written: (L15) “in the following three years”
Reviewer L 16 not sure that the sequencing provided in this paper provides information on dynamics.
Answer: Thank you for your observation, although the results did not allow us to describe the evolutionary dynamics of CHIKV, we present information that allowed us to reconstruct the dynamics of the viral spread during the outbreak in Mexico.
Reviewer: L22 only is unclear here – why only? Perhaps remove the word only?
Answer: Thank you for the suggestion, the word “only” was removed. Now it is written: (L25) “were present in our sequences”
Reviewer: L44 This reference is quite dated, and the number of cases is >10 million (ibid)
Answer: Thank you for the suggestion, the reference has been updated. Now it is written: (L52)” infected up to 10 million people”
Reviewer L201 “None of the non-synonymous mutations was found in more than one sequence “this needs rephrasing “all non-synonymous mutations were unique to each isolate”?
Answer: Thank you for the suggestion, the sentence has been rephrased. Now it is written :(L212-214) “All non-synonymous mutations were unique to each isolate”
Reviewer L207 Again “only” is used – quite unclear.
Answer: Thank you for the suggestion, the word “only” was removed. Now it is written: (L219). “Were present in our sequences”
Reviewer: L217 Not clear what a pre-analytical phylogenetic analysis might be.
Answer: Thank you for this reviewer´s observation, as a pre-processing step, sequence recombination was screened. We change the word pre-analytical for pre-processing to avoid confusion. Now it's written: “(L228-229) We did not find recombinant sequences in the pre-processing phylogenetic analysis step”.
We are grateful to the two anonymous reviewers for their precise revisions and suggestions; they have much assisted us in reanalyzing our findings with new criteria. We hope this new version is in better shape for publication, but we will entertain any further suggestions for its improvement.
Sincerely,
Mario Henry Rodríguez
Reviewer 2 Report
In this manuscript, the authors implemented the phylogeographic analysis of CHIKV sampled from patients’ serum between 2015-2016 in Mexico, aiming to investigate the spatial spread of the viruses. By sequencing three partial viral genes (np3, E1 and E2) based on amplicon-based deep sequencing, more substitutions located on these genes are found and confirmed that the circulating strains of CHIKV in these years were from the Asian genotype, more specifically Caribbean clade. The authors attempted to reconstruct the spatiotemporal dynamics of CHIKV in Mexico, which is important in the investigation and identification of hotspots of virus transmission. However, there are concerns that need to be addressed, as detailed below.
- Since the viruses are sampled from human serum. The authors should provide the ethical statement and/or study approval for this study. Regarding the data availability, the genomic data should be available when the manuscript was in the consideration of publication.
- Figure 1 is not properly displayed in the manuscript. What does the value range (1-1339) mean in each node label of the tree? the nucleotides of each sequence? If all are the same, please consider deleting them to save space and better presentation. Same case in Figure 2 (1-1354). Why are the nucleotides of sequences different for the construction of these two trees and also inconsistent with that stated in 3.2 section (the concatenated genes are 1341 nucleotides)?
- In Figure 2, the posterior probabilities of some of the main clades are not shown. Thus, in 3.3 section of Results, it is hard to agree with the inference due to the lack of supports. Phylogeographic reconstruction using discrete locations has been shown to be sensitive to spatiotemporal sampling bias. The location bias in the study seems a bit obvious before (Table 1) and after selection (Line 189-193).
- Discuss, Line 414-424, some results are probably misstated. For example, Chiapas (2014-11-15) is grouped within Clade C and why the authors talked about its spread in Clade A? And there is a possibility that the isolate Jalisco (2014-10-25) shared the common origin with that of Clade C instead of “not cluster with any of the three clades”.
- Materials and Methods, Line 97-100, it would be nice if the authors provide the result in the supplement.
- Line 107-111, the method for preparation of the library is too simple. Better to provide important information regarding the experimental design as well as index, adaptors, amplicon length, amplicon gel image, etc. as they could be useful for the studies of surveillance and inter-host mutation of CHIKV on a large scale.
- 2 section Data sets. The author should provide an overview regarding sequencing coverage, number of raw reads, filtered reads, trimmed reads, and collapsed reads etc. for each library (in supplement).
- 4 section Phylogeographic inference. The method is not clearly stated, and key parameters are not given. Line 159-161, TMRCA result is not presented in the results. Line 162-167 is not informative.
- Table S2, please explain more about this table and how it supports your conclusion?
- Figure S1, please link the abbreviation of locations on the map to the table on the right and explain what the number means?
Round 2
Reviewer 1 Report
Improved although novelty is still a bit low
2 typos in the Abstract
L13 CHIKV
L21 the Asian _viruses__were_
Reviewer 2 Report
Thank the authors for the edition. Here are the comments for this version.
- Table S2. Please provide a schematic structure about how the library look like. For example, P5-index-adaptor-locusspecificsequence-insert-adaptor-P7. If I understand correctly, the amplicons are amplified in triplex on-step PCR reaction, then the adaptors used in one reaction should be three pairs targeting NSP3, E2 and E1? Two rounds of PCR are conducted in the library preparation. One is for amplicon amplification and the other one is to add Illumina adaptor and index. Please clearly state it in the supplemental information or in the method part of the main manuscript.
- Figure S1. Selected regions studied in the study can be labelled in this figure.
- Figure S2. Please state that this is only gel image for selected serum samples if this is the case.
- Table S4 and Figure S4. Table S4 and Figure S4. I notice that the locations with more sequences included in the analysis are prone to be significant in phylogeny-trait association test. This is why I concerned that sampling bias and phylogeographic inference based on the sampling strategy. The author should discuss it in the discussion and point out the weakness. The other question is that if locations with single one sample (GUE, MEX) can be included in the test? How is one sample used for statistical analysis?
- Section 2.1, line 113-114. Please simply describe the DNA extraction method between gel and AMpure beads cleanup.
- Section 3.1, line 191-192. But the number is 253 in Table S3?
- Please put a note to explain the star symbol highlighting the substitutions.
- Figure 1. Please provide PP for the main clades, and better to label the clades like ECSA, IOL, Asian, Caribbean etc.
- Section 3.3, line 293. It is Baja California Sur (B.C.S.) instead of Baja California?
Line 304-305. I would rather include this strain in Clade B.
Line 306. Is it supposed to be Veracruz 2015-09-11 instead of Quintana Roo?
Line 307-309. These are confused words (“at the end of the clade…”). Something like “the strains from BSC and one of Nuevo Leon isolates grouped within one cluster…” would be clearer.
Line 312-313. This is what I agreed with except “same as for the first clade”.
Line 315-316. “The next internal… probability of 0.51)” removed?
Line 316-319. Colima strains are divided into three small clades. Please rephrase the description.
Line 325-326. “with the most probable root location in Quintana Roo…” It is hard to agree with it.
Line 328-330. With the high support, this strain is actually grouped in Clade C. Corresponding point in Discussion is line 436-438.
- Figure 2. Figure resolution needs to be improved.
- Discuss, line 389. Two “that”.
Line 414. “in Mexico and worldwide”. Remove worldwide?
Line 426- 429. It would be good to add one sentence of discussion for this paragraph.
Line 431-432. Four sequences from southern Chiapas grouped with Nicaragua in figure 1, while the fifth one is within another cluster.
Line 432-435. The reason is not well supported the hypothesis.
